# Maintenance Prediction through Sensing Using Hidden Markov Models—A Case Study

**Alexandre Martins** [1,2,*] **, Inácio Fonseca** [3] **, José Torres Farinha** [3,4] **, João Reis** [1,5]
**and António J. Marques Cardoso** [2]

1 EIGeS—Research Centre in Industrial Engineering, Management and Sustainability, Lusófona University, Campo Grande, 376, 1749-024 Lisboa, Portugal; p40500@ulusofona.pt
2 CISE—Electromechatronic Systems Research Centre, University of Beira Interior, Calçada Fonte do Lameiro, P-62001-001 Covilhã, Portugal; ajmc@ubi.pt
3 ISEC/IPC—Polytechnic Institute of Coimbra, 3045-093 Coimbra, Portugal; inacio@isec.pt (I.F.); tfarinha@isec.pt (J.T.F.)
4 CEMMPRE—Centre for Mechanical Engineering, Materials and Processes, University of Coimbra, 3030-788 Coimbra, Portugal
5 GOVCOPP—Department of Economics, Management, Industrial Engineering and Tourism, Campus Universitário de Santiago, Aveiro University, 3810-193 Aveiro, Portugal
* Correspondence: alex_bmts@hotmail.com

**Abstract:** The availability maximization is a goal for any organization because the equipment downtime implies high non-production costs and, additionally, the abnormal stopping and restarting usually imply loss of product's quality. In this way, a method for predicting the equipment's health state is vital to maintain the production flow as well as to plan maintenance intervention strategies. This paper presents a maintenance prediction approach based on sensing data managed by Hidden Markov Models (HMM). To do so, a diagnosis of drying presses in a pulp industry is used as case study, which is done based on data collected every minute for three years and ten months. This paper presents an approach to manage a multivariate analysis, in this case merging the values of sensors, and optimizing the observable states to insert into a HMM model, which permits to identify three hidden states that characterize the equipment's health state: "Proper Function", "Alert state", and "Equipment Failure". The research described in this paper demonstrates how an equipment health diagnosis can be made using the HMM, through the collection of observations from various sensors, without information of machine failures occurrences. The approach developed demonstrated to be robust, even the complexity of the system, having the potential to be generalized to any other type of equipment.

**Keywords:** Hidden Markov Models; industrial sensors; condition-based maintenance; big data; cluster analysis; principal component analysis





## 1. Introduction

Sensors are currently one of the largest sources of data, being responsible for making a direct connection between a physical phenomenon and a data acquisition system, converting signals from several types of variables (mechanical, chemical, etc.) into electrical signals. Thus, sensors are responsible for translating the equipment condition, giving outputs that can be seen as observable states. In other words, they provide data that, after being studied, can provide very useful information for companies. One of the ways in which companies maintain competitiveness and customer satisfaction is through the application of sensing, as this will allow them to carry out Condition-Based Maintenance (CBM). This is aimed at the increasing of profits, namely, due to the non-existence of unexpected stoppages in production. For instance, according to Pais et al. [1], Data Mining and Artificial Intelligence (AI) can contribute to safeguard the company's profits and the safety of people and property.

This work aims to demonstrate how it is possible to optimize the data collected from the raw signal given by the sensor and, through the transformation of observable states made by the HMM, how the equipment diagnosis can be obtained. The case study aims to diagnose the state of health of a drying press in a pulp industry, where the section task is to remove water, mainly through pressing. Further, the drying press in a pulp industry is an equipment where the occurrence of failure is harmful to the production chain, which can lead to loss of time and high costs for the company. Therefore, to diagnose this equipment, we have collected data with six sensors, which are coupled to the equipment and are responsible for measuring the current intensity, the level of the hydraulic unit, the torque, the VAT pressure, the speed of rotation, and temperature of the hydraulic unit, in order to characterize the health status of the equipment. This will be done by joining the data from the six sensors and optimizing the observable states, which will later be inserted into an HMM model, allowing the equipment to be diagnosed. The optimization of observable states goes through four phases: data preparation; features generation; Principal Components Analysis (PCA), and Clustering. With this procedure it is possible to extract more information from the signals, to reduce the computational load, and to join the observations of the six sensors, allowing the creation of only one HMM to diagnose the equipment. The objective is to characterize the health of the equipment in three states over the time of data collection: "Adequate Functioning", "Alert state", and "Equipment failure".

A theoretical basis is also made on the entire methodology used to carry out the equipment diagnosis in order to demonstrate how the HMM can be applied to assist in Condition-Based Maintenance.

As Rodrigues et al. [2] stated, there has been extensive research on maintenance optimization that has also been an important trend in the area of optimization based on maintenance simulation. The authors state that research in this context focuses on the goal of finding the best maintenance policy while minimizing the overall company costs. Like Mateus et al. [3] state, "it was clear that a well maintained property will have a longer useful life with a greater return for the organization". The main objective of this type of maintenance is to use adequate sensor signals and monitoring techniques to identify and predict the health status of machines, in order to reduce the economic loss due to degradation or failure [4]. Another aspect to consider when talking about data collection with sensors is their reliability [5]. As noted by Taylor et al. [6] and by Bunks, Mccarthy, and Al-Ani [7], a prerequisite for the implementation of effective CBM practices in the industry is an effective diagnosis and prognosis.

According to Kamlu and Laxmi [8], the CBM decision-making process based on the information obtained from the HMM is adequate to recognize the condition of the equipment. The HHM is a double stochastic process that can define, through probability, the equipment diagnosis, that is, how the machine can go from a good working state to a bad working state and what the statistics are in each state.

Bunks et al. [7] explain that the HMM has two very useful features when it comes to monitoring CBM equipment: first, they refer to the existence of computationally efficient methods to calculate probabilities (an important feature, as it promises to be an efficient tool for signal processing that can be economically implemented), and second, they mention the existence of efficient techniques that can be used to identify the system health with HMM, that is, they can be used to build equipment models based on data, aiming to identify specific characteristics in the data to be used as health indicators.

There are several examples of HMM-based approaches to monitor failure detection [4,6–15]: Yu [4] uses an unsupervised online learning scheme, where an Adaptive Hidden Markov Model (AHMM) is used to learn online the dynamic changes in the health of the equipment; Taylor et al. [6] present a method where several HMMs are used to represent a health state of a metal cutting tool; BUNKS et al. [7] show, in addition to why the HMM is a model for equipment diagnostics, its use with vibration data to diagnose helicopter gears; Arpaia et al. [10] use the HMM to detect failures for fluid machines without

adequate a priori information about failure conditions; Ocak and Lopar [12] present a bearing failure detection and diagnosis scheme based on vibration signals; Xinmin et al. [13] focus on the analysis of the bearing failure diagnosis model concentrated in an HMM model, and compare it with other approaches, namely Multilayer Perceptron (MLP) detection techniques; and Simões et al. [14] present an extensive description of the state of the art of the HMM, also describing how this model can increase the quality of the assessment of Diesel engine conditions and the efficiency of maintenance planning.

In relation to the study carried out, this paper demonstrates how by performing a multivariate analysis using several different sensors and with only one HMM it is possible to obtain a diagnosis of the equipment, in this case, a drying press. This approach is used instead of using one HMM for each sensor or multiple HMMs for each sensor, where each HMM represents an equipment health state. In this paper, the steps to be followed to optimize the HMM inputs will be demonstrated, and it will also be shown that through features generation and extracting features (using PCA), it is possible to merge the various sensors.

The paper starts with an introduction to the general theme in Section 1. In Section 2, the methodology used to optimize observable states for HMM is explained. Section 3 presents a theoretical framework for each of the steps, including the HMM model itself. Section 4 explains the case study and demonstrates all the steps taken until the equipment is diagnosed. Section 5 discusses the results and explains ideas for future work. Finally, in Section 6, the conclusions of the study are presented.

## 2. Theoretically Background

This section aims to provide knowledge of the literature review on the topics covered in this paper. That is, all the processes used to "optimize" the observable states, as well as the HMM model used to diagnose the equipment.

### 2.1. Data Preparation

According to Yin et al. [16], since the beginning of the 1980s data mining has gained increasingly more attention as a means of obtaining knowledge.

The preprocessing of raw data facilitates the stabilization of the mean and variance and aims to provide a structural, reliable, and integrated data source, and it is a very critical and complex step, which allows guaranteeing reasonable results, whether the analysis is concerned with exploratory data mining, classification, or construction of a good and robust prediction model [17–21].

Data cleaning and preparation consume approximately 80% of the total data engineering effort, as it is a process that may require many transformations and be repeated many times [21–23].

Through good data cleaning, it is possible to improve performance and quality [17,24,25], as well as to isolate characteristics of interest and eliminate elements that "bother" the theoretical models [26].

Data Transformation

Data formats can be transformed to meet the premises of a statistical inference procedure or to improve interoperability [27]. Data normalization is the process of transforming raw data values into another format with properties better suited for modeling and analysis. For a variable X represented by a vector $\{x_1, x_2, x_3, \ldots, x_n\}$., we can use Z-score normalization (also known as standardization).

Z-score normalization is a normalization method that transforms not only the magnitude of the data, but also the dispersion. It will convert a variable range with some mathematical heuristics, allowing all variables to have the same range. This patterning of values is not essential in machine learning algorithms, but it can make patterns in the data more visible [23]. Z-score normalization overcomes the problem of variables with different

units, as it transforms the variables; therefore, they are centered on 0 with a standard deviation of 1 [21]. Each attribute can be transformed using Equation (1):

$$x'_i = \frac{x_i - \overline{X}}{std_{dev}(X)} \tag{1}$$

where

- $x_i$ is the Z-score value of $x$;
- $\overline{x}$ is the row mean of $x$;
- $std$ is the standard deviation given by

$$std_{dev}(\overline{x}) = \sqrt{\frac{1}{n-1} \sum_{i=1}^{n} (x_i - \overline{x})^2} \tag{2}$$

### 2.2. Features Generation

The principle of predictive Feature Generation (FG) is used to maximize the exploitation of information generated exclusively from time and process data, with compact and informative representations of the obtained data [27,28].

According to Silva and Leong [29], Machine Learning (ML) algorithms can be seen as techniques to derive one or more hypotheses from a set of observations, thus being one of the ways to improve input (observations) and output (hypothesis) in selecting resources that maximize the performance of architecture.

If the transformation is chosen correctly, the transformation features can display high-information packing properties compared to the original input samples.

In summary, FG is of paramount importance in any pattern recognition task, whose objective is to discover compact and informative representations of the obtained data [30].

### 2.3. Principal Component Analysis (PCA)

Dimensionality Reduction is a technique to take a high-dimensional dataset (data objects with many features/attributes), replacing it with a much smaller dimensional dataset, preserving the similarities between the data objects [31].

Principal Components (PC) were first found by Pearson [32] and gained consideration by the statistical community when Hotelling [33] proposed them as an estimate of linear combinations of a set of random variables that retained the greatest possible variance.

Roy et al. [27], Anowar et al. [34], and Warmenhoven et al. [35] argue that it is an unsupervised linear transformation algorithm and that it uses a statistical procedure through an orthogonal transformation to map a set of observations of possibly correlated variables into a set of linear values of uncorrelated variables called Principal Components.

In short (Table 1), PCA is a method that seeks to find linear combinations of predictors, known as principal components (PCs), that capture the greatest possible variance [36]. The first PC is defined as the linear combination of predictors that captures the greatest variability of all possible linear combinations. Subsequent PCs are derived, so these linear combinations capture the greatest remaining variability, although they are not correlated with all previous PCs.

**Table 1.** PCA algorithm steps.

| ALGORITHM—PCA |
| --- |
| **INPUT: $X \in R^{n*d}$** |
| **OUTPUT: Y $\in R^{n*k}$** |
| **1: CONSTRUCT THE COVARIANCE MATRIX ($X.X^t$)** |
| **2: APPLY LINEAR EIGEN DECOMPOSITION TO $X.(X^T)$ TO OBTAIN EIGEN VALUES AND VECTORS;** |
| **3: SORT EIGEN VALUES IN DECREASING ORDER TO SORT EIGEN VECTORS** |
| **4: BUILD MATRIX $W(D*k)$ WITH $K$ TOP EIGEN VECTORS** |
| **5: TRANSFORM $X$ USING $W$ TO OBTAIN THE NEW SUBSPACE $Y=X.W$** |

*2.4. Clustering K-Means*

Clustering is a fundamental unsupervised data mining technique that is used in a wide variety of areas for data analysis [37], where cluster analysis is done by multivariate statistical methods and the objective is to identify clusters of objects within the data, maximizing the homogeneity within each cluster and the heterogeneity between the different clusters [18,37–41].

The k-means algorithm was first proposed by Stuart Lloyd in 1957 as a technique for pulse code modulation, with a more efficient version being proposed and published in Fortran by Hartigan and Wong in 1975/79 [42,43].

It is a numerical, unsupervised, iterative, and evolutionary algorithm, which aims to find the positions of the clusters that minimize the distance from the data points to the clusters, it then partitions the *n* observations into *k* clusters in which the observation belongs to the cluster with the most average next [42,44].

Each cluster in the partition is defined by its member objects and its centroid, with the centroid being the point at which the sum of the distances of all objects in that cluster is minimized [43]. Then, K-means finds a centroid to correlate the data to a specific group, and the process of estimating the data belonging to a given cluster is carried out iteratively until the convergence condition is satisfied [39].

To define the cluster number *k*, the Silhouette method can be used, because, according to Menardi [45], the diagnostic used to assess the quality of a partition must be consistent with the clustering method adopted to produce that partition, being Silhouette parsing a method that fits well with the K-means Clustering algorithm.

A detailed explanation of this topic can be found in [37,45,46]. Peralta and Saeys [47] summarize by saying that the Silhouette Index (SI) is calculated for each resource and measures it through the ratio between intra-cluster and inter-cluster dissimilarity $a(k)$ and $b(k)$, respectively, as shown in Equation (3), generating a value between −1 and 1.

$$S(k) = \frac{b(k) - a(k)}{max\{a(k), b(k)\}} \tag{3}$$

*2.5. Hidden Markov Models (HMM)*

Hidden Markov Models (HMMs) started to be introduced around the 1970s, with the publications of Baum [48] and Baum & Petrie [49], and later became quite popular in the late 1980s with the contribution of Rabiner [50]. Currently, the HMM is a statistical modeling tool that has become popular in several areas, such as speech processing, DNA recognition, weather forecast, machine maintenance, pattern identification, and health monitoring.

HMMs are based on a doubly stochastic process, in which an underlying stochastic process that develops as a Markov chain produces an unobservable ("hidden") state, which can be inferred only through another set of stochastic processes [51]. Therefore, an HMM is a stochastic technique for modeling signals that evolve through a finite number of states. States are considered hidden and responsible for producing observations [4,6].

A typical notation used for HMM is

- *N*—Number of states

- *M*—Number of observations
- *T*—Observation sequence length
- Sequence of observations up to T is presented as:
- $O_1, O_2, O_3, \ldots, O_T$ with $O_t \in \{V_1, V_2, V_3, \ldots, V_M\}$
- Corresponding status sequence is displayed as:
  - ○ $Q_T = \{q_1, q_2, q_3, \ldots, q_T\}$ with $q_t \in \{S_1, S_2, S_3, \ldots, S_N\}$
- $\lambda = (A, B, \pi)$ – is a representation of an HMM model *A*—Transition probability matrix between hidden states

$$A = \{a_{ij}\}, 1 \leq i, j \leq N \tag{4}$$

where:
- $a_{ij} = P$ and $q_t$ represents the hidden state at time *t*
- *B*—Emission Probability Matrix

$$B = \{b_j(k)\}(1 \leq j \leq N) \tag{5}$$

- $\pi$—initial state probability distribution

$$\pi_i = P(q_t = S_i) \tag{6}$$

where the observations are issued from each state according to the probability distribution

$$b_j(k) = P(O_k \vee q_t = S_j), \text{ it's} \tag{7}$$

An HMM has several hidden states (*N*), an initial probability value ($\pi$) for each state, a transition probability matrix (*A*) indicating the probability of transition from one hidden state to another, and a probability matrix of transition (*B*) which indicates the probability of an observation given a certain hidden state. The sum of all initial probabilities must be equal to 1, as well as the sum of all elements in a row in the transition and issue probability matrix [50].

According to Rabiner [50], there are three basic HMM problems:

*Problem 1*—This is an evaluation problem, where, given a sequence of observations $O = \{O_1; O_2; O_3; \ldots; O_T\}$ and the model $\lambda = (A, B, \pi)$, it allows to efficiently to calculate the associated probability to the $P(O \vee \lambda)$ emission sequence.

*Problem 2*—This is a decoding problem, which consists of finding the most likely sequence of hidden states given the sequence of observed emissions $O = \{O_1; O_2; O_3; \ldots; O_T\}$ and the model $\lambda$, namely, how to find a corresponding sequence of states $S = \{S_1; S_2; S_3; \ldots; S_T\}$.

*Problem 3*—This is about to know a database of sequences and how to adjust the parameters of the model $\lambda = (A, B, \pi)$ in order to maximize $P(O \vee \lambda)$.

➢ The solution for Problem 1

To solve the first problem, the forward and backward algorithms must be used. In this way, it is possible to find the forward variable $\alpha_t(i)$, defined as

$$\alpha_T(i) = P(O_1, O_2, O_3, \ldots, O_T, q_T = S_i \vee \lambda) \tag{8}$$

This variable will give the probability of the observation sequence, $O_1, O_2, O_3, \ldots, O_t$ and state $S_i$ at time *t* given the model $\lambda$.

Through the forward and backward procedure, we also obtain the backward variable $\beta_t(i)$, defined as

$$\beta_t(i) = P(O_{t+1}, O_{t+2}, O_{t+3}, \ldots, O_T \vee S_t = q_i, \lambda) \tag{9}$$

This variable gives the final probability of the observation sequence from $t + 1$, given the state $S_i$ at time *t* and the model $\lambda$. This variable is not necessary for this first problem, but it will be useful to solve *Problem 3*.

➢ The solution for Problem 2

This problem aims to find the best sequence of hidden states that best fits the observed states. This is done using the Viterbi algorithm.

➢   The solution for Problem 3

The third problem is to adjust the HMM parameters to maximize the probability of the observation sequence. This is done using the Baum–Welch algorithm.

The Baum–Welch algorithm (or Baum–Welch expectation-maximization algorithm) makes use of both the direct variable $\alpha_T(i)$ and the regressive variable $\beta_t(i)$ when determining updated parameters for the HMM. Because of this, the Baum–Welch algorithm is also known as the Forward-Backward algorithm [11].

As Yu [4] refers, three basic algorithms are used in HMM: the Forward-Backward procedure, the Baum–Welch algorithm, and the Viterbi algorithm, all of which are used for learning and recognizing the parameters of the model $\lambda = (A, B, \pi)$.

## 3. Global Framework

Rabiner [50] states that "Real-world processes generally produce observable outputs which can be characterized as signals." These observable states will be used as input to the HMM model which will later give us the hidden states which, in turn, will correspond to the equipment state. To improve these inputs to the model, a sequence of methodologies is performed to perform the "optimization" of the observable states (Figure 1).

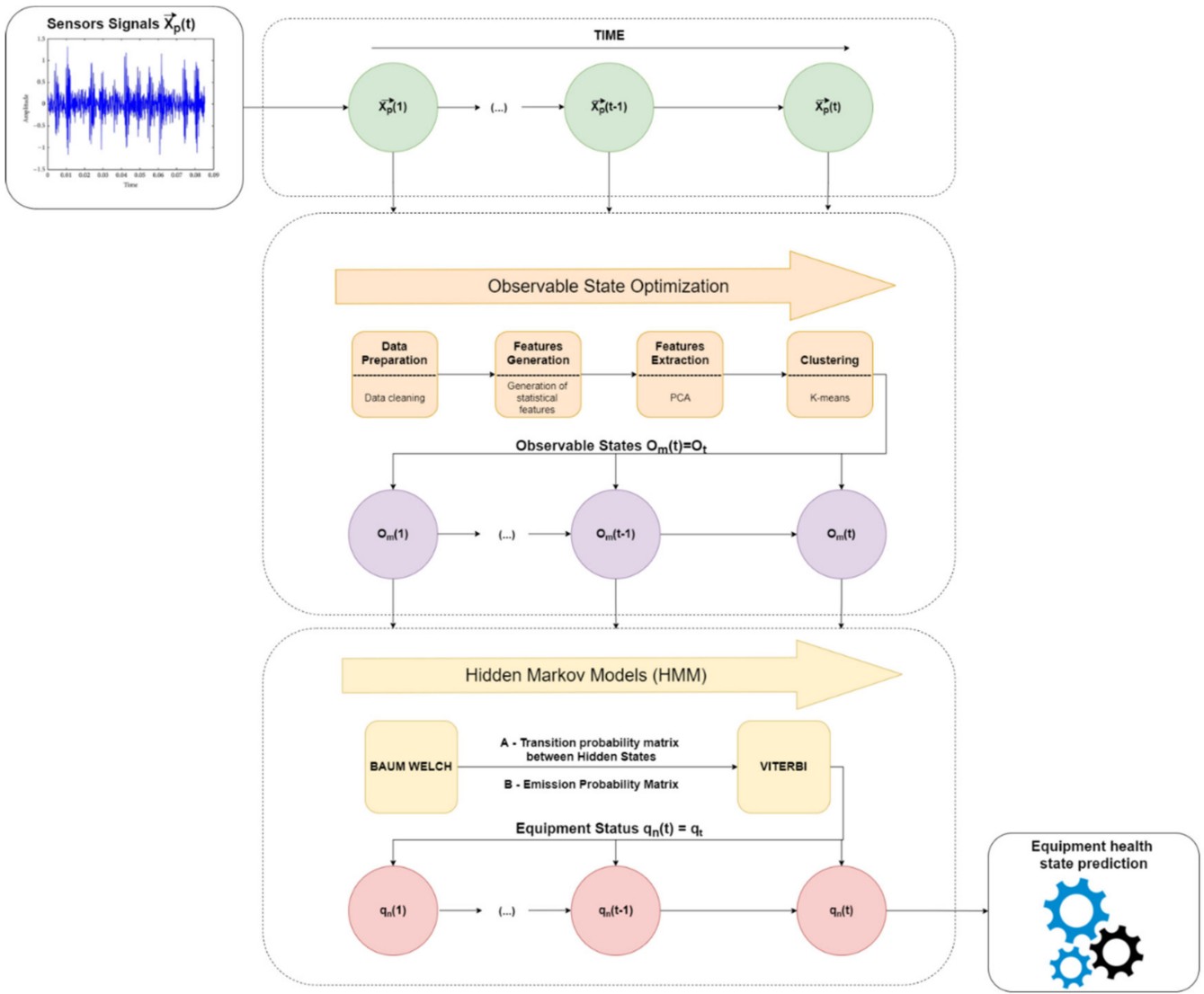

**Figure 1.** Methodology for observable states optimization to insert into the HMM.

As we can see in Figure 1, the sensor data will be modified to obtain better algorithm efficiency, both in calculation time and global performance.

First, the data are collected by sensors over time, which, later, are managed aiming to optimize the observable states. After data collection, these go to the preprocessing stage, where a data preparation is carried on, to improve the quality of the data set.

At the end of data preparation, the data are grouped in time intervals and, after, replaced by statistical values representative of each interval. Based on this generation of characteristics, it is possible to extract more information from the data, as the statistical resources, based on the time domain, providing good performance to characterize trends and changes in data [52].

The data are then normalized and inserted into a dimensional reduction model using Principal Component Analysis (PCA). Sizing features through Z-Score standardization is an important preprocessing step to resize features according to a standard normal distribution, with a mean zero and a standard deviation one. In our case, it is important to use the Z-Score standardization to implement PCA method because we are interested in the components that maximize the variance. If one feature varies less than another, because of their respective scales, the PCA defines the direction of the maximum variation that more closely matches the axis, which varies more if those features are unscaled. It is important to standardize the data to make the algorithm more sensitive to the changes that may occur in different measurements. Therefore, standardization is performed because this model requires resources to be on the same scale aiming to find directions that maximize the variation. In this process, we will only obtain the principal components (PCs) for the study and reduction of the dimensionality of the dataset. The observations are now defined in a new space, redefining the axes through the PCs instead of the original variables. In other words, the characteristics of the data will be reduced to reduce the computational load. In this study, the PCA was chosen; it is a very consolidated and easy method to use. It is an unsupervised model that permits to find the most significant coordinate system of the data to find the strongest characteristics of the samples. Additionally, it is not time-consuming, it reduces well the overfitting, and can be used as a noise removal and data compression technique [27,34,38,53]. The elimination of the large part of the experimental noise is possible—the noise originates random errors, which are not correlated with the information contained in the data matrix—improving the numerical stability of the model [36]. Finally, a clustering is performed, which will group similar data and differentiate from data from other data groups. In other words, clustering will group the observations as homogeneously as possible within each cluster and as heterogeneously as possible among the different clusters. In this way, observations of the equipment in good working order will be grouped in a cluster, observations of the equipment in failure will be grouped in another, and so on. Each cluster formed will, in turn, be used as an observable state that will enter the HMM. The method used in this study will be Clustering k-means, which does the grouping using Euclidean metrics. k-means is the algorithm chosen to carry out our study; because it is a model that uses the unsupervised learning method, it has a fast enough convergence speed and does not require large computational resources; it is an easy algorithm to implement and apply, even to large data sets [33,34].

In addition, it is an algorithm that can be used after refocusing the data from the source of the metrics framework involved in the center of the cloud (process done by the PCA). This causes the data to be centered in relation to an origin where it is positioned, being relatively equidistant from the center and positioned in the several quadrants in the $Rp$ space ($p$ being the PCA's Principal Components), which will allow a good performance of the K-means.

Finally, the HMM will give us, through the hidden states, the diagnosis of the equipment.

## 4. Case Study

This case study focuses on the fault diagnosis status of equipment (i.e., a drying system press) in a Portuguese paper pulp industry. The objective is, through the data provided by the company, to be able to make a diagnosis of the equipment state in order to know if it is in "Good Operation", "Alert Status", or "Faulty Equipment", using the HMM model. This case study's specification adds value about evaluating the equipment diagnosis, namely, because it does not need a priori information concerning the failure conditions. To carry out the study, the MatLab software and its functions were used.

### 4.1. Data Preparation

The data used for the study were collected by six sensors (responsible for measuring current intensity, hydraulic unit level, torque, VAT pressure, rotation speed, and hydraulic unit temperature), which collect an observation every minute. The data collected are from August 2017 to October 2020. Thus, the data used were acquired over three years and three months (Table 2).

**Table 2.** Used sensors: Number of data collected.

| Equipment | Sensors | Metric | N° of Data by Day (Each Sensor) | N° of Data by Year (Each Sensor) | TOTAL of Data (Each Sensor) |
|---|---|---|---|---|---|
| Drying Presses | Sensor−1 | Hydraulic unit Temperature | 1440 | $1440 * 365 =$ $525600$ | $(1440 * 153) +$ $(525600 * 2) +$ $(1440 * 305) =$ |
| | Sensor−2 | Hydraulic unit level | | | |
| | Sensor−3 | Rotation speed Press | | | |
| | Sensor−4 | Torque Press | | | |
| | Sensor−5 | Current intensity with endless extraction drive | | | |
| | Sensor−6 | VAT Pressure vat press | | | |

Table 2 shows that there is a dataset with 10,264,320 observations. However, as we referred above in the theoretical framework, there are always errors in sensor data collection and, therefore, not all data in this dataset are valid. According to Van Den Broeck et al. [54], data cleansing should be based on knowledge of technical errors and expected ranges of normal values. As such, through a data collection confidence index, it was possible to verify whether the collected values were valid or not. This confidence index is a protection of the data acquisition system, which through just two values (0 or 100) indicates the quality of the data. Regarding the implementation of this system, there are not many details as it was installed by a tertiary company. If the index had a value of 100, it meant that the data had been well collected and they were valid, and when the index had a value of zero, it meant that something had happened with the data collection and, as such, they were considered invalid. Therefore, it was decided to eliminate all points whose confidence index was equal to zero.

In addition to this data cleaning, it was also decided to make an imputation of the total mean of the data signal, whose values from the current intensity, torque, VAT pressure, and rotation speed sensors were below a certain threshold. In this way, we guarantee that we can remove the equipment shutdown and to replace them by the respective average of each signal. Therefore, the equipment shutdowns will not affect the prediction of the model. The value imputation was performed, which is the most used, according to the most usual referred in literature, as can be seen, for example, in [23,31,55]; the most general opinion supports this approach when the data of a variable is missing. Based on this approach,

some care was considered: when a slice was replaced in one of the signals, the others were all replaced by their respective mean, increasing the integrity of the data set. No further data cleaning is performed as the following data processing models themselves complement the "filtering" of the dataset.

In Figure 2, we can graphically compare the data set with all the values, with the valid values and without pulp drying press downtimes.

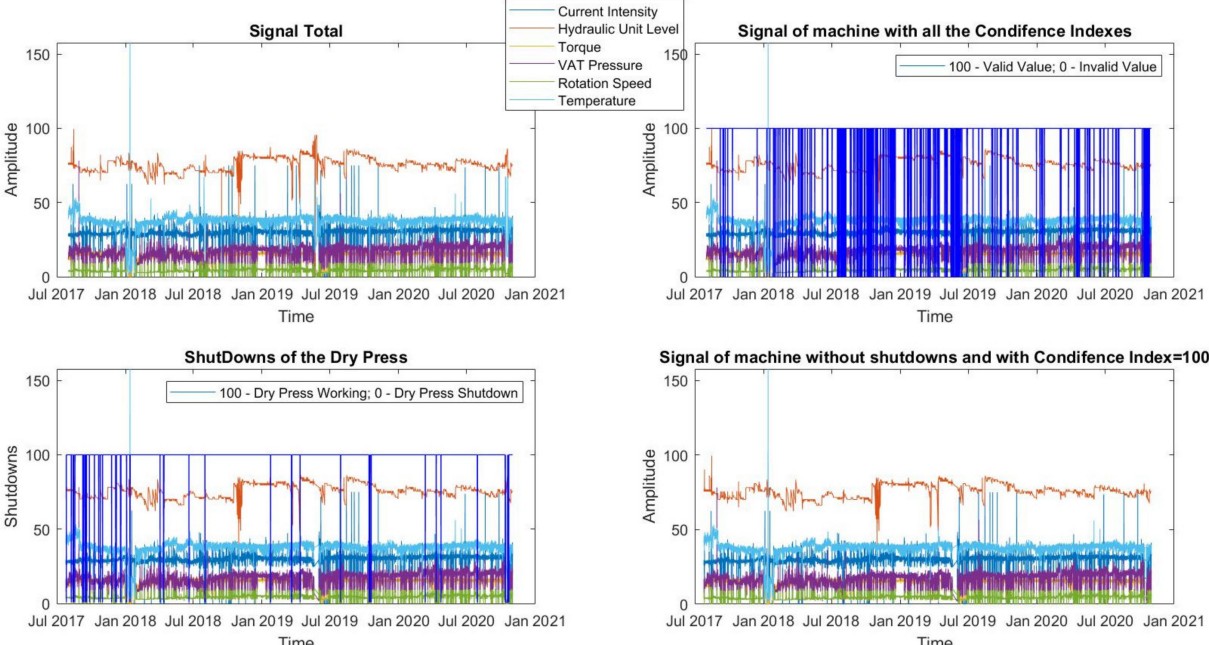

**Figure 2.** Data cleaning of the signals with through Confidence Indexes and shutdowns of the equipment.

Through the confidence index, 69,840 data were eliminated, leaving a data set of 1,640,880 per sensor. Regarding equipment shutdowns, these were replaced by the average of each signal and, therefore, the value of 1,640,880 remained unchanged.

As is well known, outliers can be classified into two different subgroups: natural and interesting ones, and those caused by defective instruments. The first group will contribute with data that may improve the analysis or construction of the subsequent model, while the second will contribute with errors that make the information extraction less accurate [21,23,56]. Therefore, in this case, as we are carrying out a study of the failure diagnosis of drying presses, we chose not to remove what could be the outliers, as in this case, because these values may represent equipment malfunctions and, by consequence, an invalid number.

### 4.2. Features Generation

Six-hour time windows were created to represent four shifts of the equipment's workday, where each window has 360 data. Thus, 4558 time windows were additionally created with 360 data each. The objective is to create statistical characteristics of each window to better characterize the signal and to make a dimensional reduction of the data. Ten different characteristics were obtained from each data window to better define the signal and understand its behavior over time. The characteristics generated were 1. Average; 2. Standard deviation; 3. Variance; 4. Kurtosis; 5. Skewness; 6. Coefficient of variation; 7. Maximum; 8. Minimum; 9. Mode; 10. Median. These 10 characteristics were chosen in order to have a limit of resources for the study to verify if they are sufficient to obtain a diagnosis. The features are all performed in the time domain, as none of the variables need frequency analysis. Furthermore, as stated in [52], "statistical resources based on the time domain provide a high performance to characterize trends and changes

in signals". The corresponding equations of the considered set of characteristics can be found in the references [52,57]. In Figure 3, we can see the features generated for each variable.

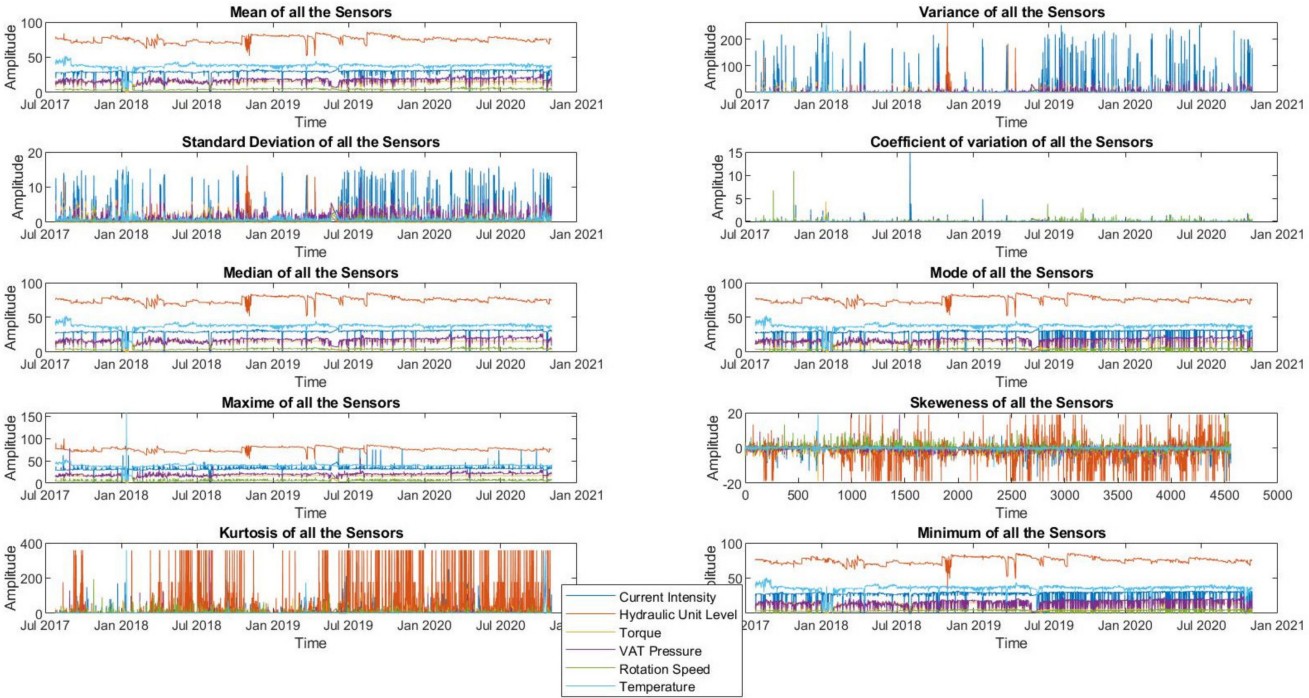

**Figure 3.** All features for each one of the signals sensors.

Based on these 10 characteristics of each sensor, 60 variables were generated from a total of 4558 data. That is, we are left with 4558 mean values for each sensor, 4558 variance values for each sensor, etc. Therefore, we get an array of a dimension of 4558 ∗ 60.

### 4.3. PCA

From the matrix built above, we made a feature extraction and reduced the size of the matrix caused by feature generation using the Principal Component Analysis (PCA) tool for the study of the principal components. From the matrix 4558 × 60, a reduction was made, starting to have a matrix with only ten main components, 4558 × 10. As already described, the first dimensions are those that contain the greatest variation of the data. In Table 3, taken from the MatLab PCA function, we can see the percentage variability of each of the main components and how it decreases in each PCA.

**Table 3.** Percentage variability explained by the main components.

| Principal Components | % of Preserved Data |
|---|---|
| 1 | 26.8299362662924 |
| 2 | 11.3444018431582 |
| 3 | 10.5209646496898 |
| 4 | 7.43400606597570 |
| 5 | 6.75048105033166 |
| ( . . . ) | ( . . . ) |

Therefore, the top ten components were chosen as this number of components preserves ~82% of the data. For a graphical analysis of the variability of the data on each PCA, we used a Pareto diagram (Figure 4).

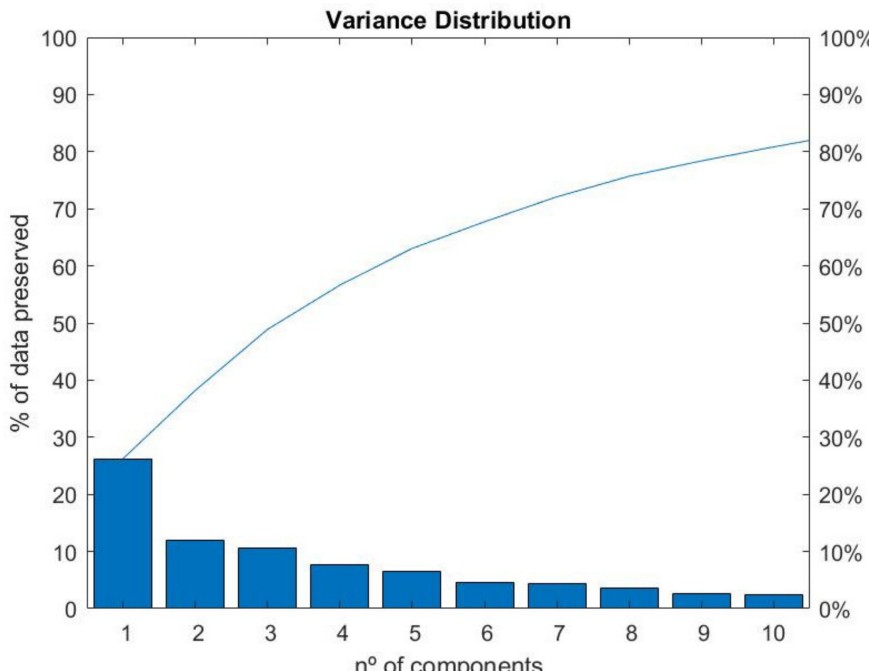

**Figure 4.** Pareto diagram to represent the percentage of data preserved in each Principal Component.

In addition to the feature extraction through PCA, a study can still be done where we can identify which features contributed most to each of the main components, through the coefficients of the main component. This can be very well represented using a heat map represented in Figure 5.

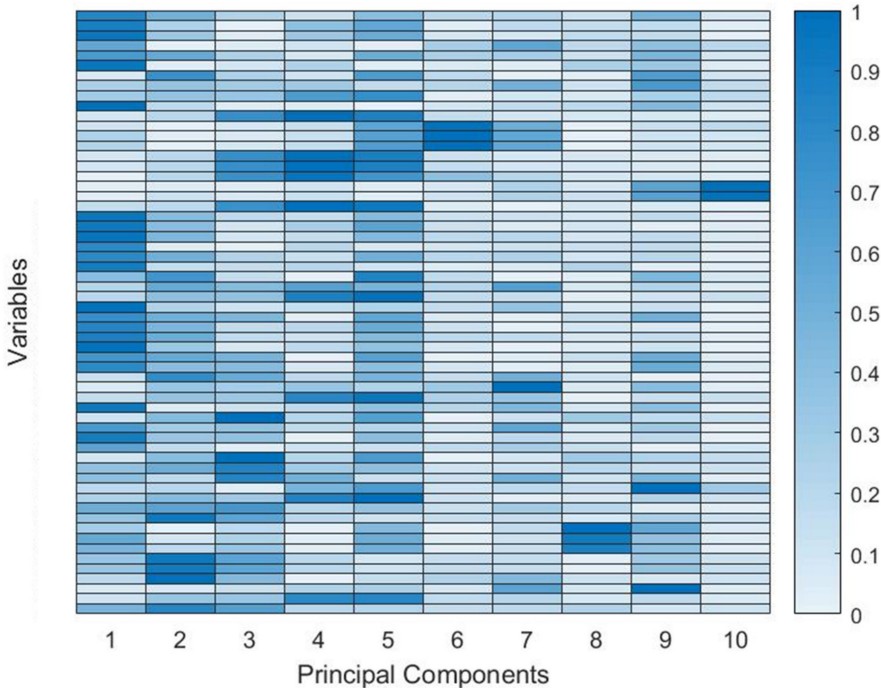

**Figure 5.** Contribution of each of the features in each of the Principal Components.

In the coefficient matrix, each coefficient column contains coefficients for a principal component and the columns are in descending order of component variance. Therefore, we represented in the lines the 60 variables extracted for each one of the sensors and, the 10 Main Components were represented in the columns. Therefore, the first 10 lines will

represent all the features extracted for the first sensor and so on, respecting the order of sensors and features described above. In this way, it is possible to verify which variables most contribute to each of the PCA, and the ones that contribute the most are the darker blue ones. As PCA number one is the one with the most variability of data, we can deduce that the features that most influence this component are the most important.

### 4.4. Clustering

Through the matrix points extracted from the PCA, a grouping was made to identify the groups that most identify themselves. Each group that identifies itself is seen as an observation of the equipment; in other words, an observation that matches a normal data equipment behavior will be grouped into a cluster that groups observations with that behavior and malfunction characteristics, that will be grouped into another cluster, and so on. The algorithm used to perform Clustering is K-means, therefore, it is necessary to define, a priori, the number of clusters that will be performed. This criterion will be done through a silhouette analysis where the silhouette criterion is used to assess the ideal number of clusters; for these data, it was found that the ideal would be three clusters (Figure 6).

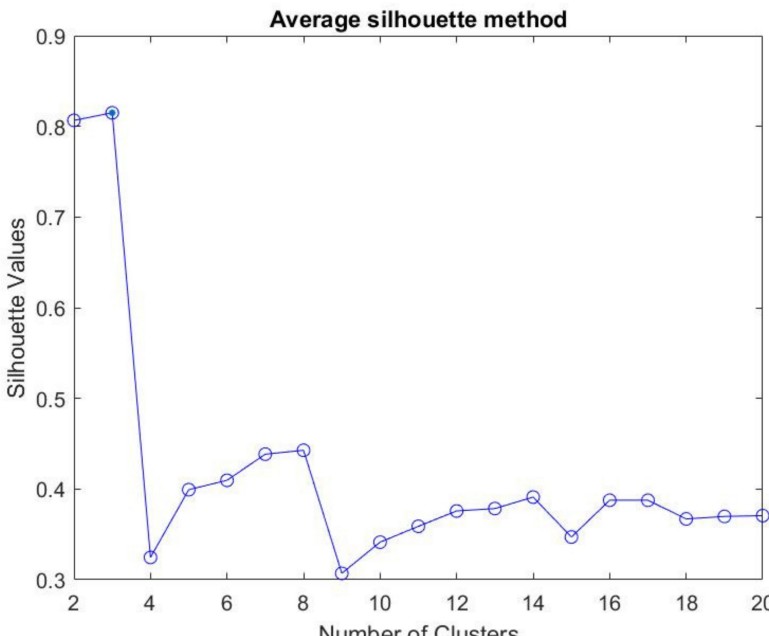

**Figure 6.** Identification of the ideal number of clusters for the sample under study through silhouette analysis.

Each cluster will then be classified as the observable state in the study period. A descending ordering of clusters was also done. In this way, we guarantee that cluster 1 is the one with the most points, cluster 2 is the second cluster with the most points, and so on. Based on this, cluster three corresponds to the observation with the fewest points and is what happens less frequently and, therefore, the rarest to happen, which leads to deduce that an observation occurs when the equipment is not in good condition (this will be demonstrated in the HMM section). In Figure 7, we can see how each cluster looks over time. As each cluster is in an observable state of the equipment, we can look at the graph and see the development of the observable states of the equipment over the study period.

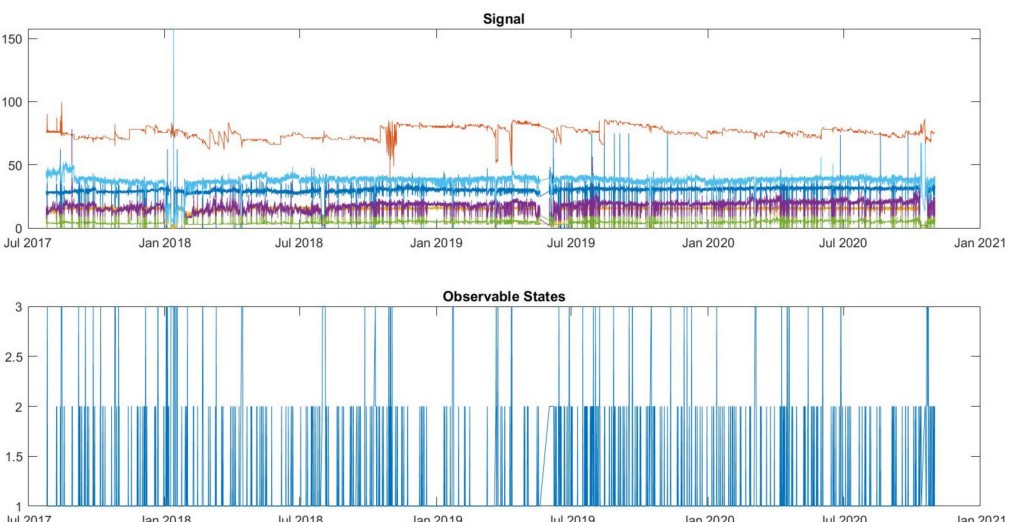

**Figure 7.** Observable states of equipment over time with K-means clustering.

In order to verify if the groups chosen by K-means make sense, we can compare the cluster development with the direct observations of the reading equipment. As we saw above, cluster 3 will be the most unlikely to happen, and it should be represented when there are observations in the less common data, as we can see in the Figure 7, that is what is happening. This leads to the inference that K-means did a good data grouping.

*4.5. HMM—Equipment Diagnostics*

In the HMM model, there are observable states (defined in the previous section) and hidden states (equipment condition). For now, we have chosen three states to characterize the equipment health status: the 1st representing the good functioning of the equipment, the 2nd identifying an alert state, and the 3rd state representing the equipment malfunction.

The accuracy of data classification used by the HMM is nothing more than the quantity of correctly validated estimates for all types of events or status classes of a system divided by the total monitoring of the condition of a system, after mounting the model [14]. Thus, through the outputs taken from Clustering, the data were divided into training data and test data to validate the model. Thus, through the outputs taken from Clustering, the data were divided into training data and test data to assess the model. According to the common practice in machine learning, 70% of the data were used to train the HMM model and 30% were used to test the performance of the system in unknown data. Considering that the data are actual time series, the first 70% of the samples were used for training and the latter 30% were used for testing.

Then, through the observable training states, based on the Baum–Welch algorithm, we were able to train the HMM to obtain the transition and emission matrices:

Transition matrix:

$$
\begin{bmatrix}
Stage & State1 & State2 & State3 \\
State1 & 0.9113 & 0.0857 & 0.0031 \\
State2 & 0.5080 & 0.4627 & 0.0294 \\
State3 & 0.0615 & 0.3291 & 0.6094
\end{bmatrix}
$$

Emission Matrix:

$$
\begin{bmatrix}
Stage & Obs1 & Obs2 & Obs3 \\
State1 & 0.9892 & 0.0108 & 0.0000 \\
State2 & 0.3133 & 0.6431 & 0.0436 \\
State3 & 0.0000 & 0.0000 & 1.0000
\end{bmatrix}
$$

Note that the division of data into training data and test data was done in a temporal way, where 70% of training data goes from August 2017 to mid-October 2019, with the remaining months for 30% of test data. In order to find out if this was a good way to divide the data, a comparison was made between the matrices trained by the training data and the matrices trained with the totality of data. Through the Root Mean Square Error metric (Equation (10)), we obtained the following results:

$$\text{RMSE} = \sqrt{\frac{1}{n} \sum_{ij}^{n} \left(a_{Data_{Total}}(i,j) - a_{Data_{Train70\%}}(i,j)\right)^2} \tag{10}$$

$$RMSE_{Transition=0.0709}$$

$$RMSE_{Emission=0.0289}$$

Through this metric, we verified that the division performed in a temporal way was a good choice and, from here, we can also take the model to be trained that does not need all the data.

Having trained the model, through the obtained matrices, it is now possible to generate observable states and to compare them with 30% of the test data. Then, a comparison is made between the sequence of observable states calculated by the model already trained, and the sequence of observable states that actually exit Clustering (test data). Through the test data, it becomes possible to calculate the probability, $y$, of such sequences overlapping [58]. In other words, based on the collected data, it is possible to specify the probability of the estimated states corresponding to the real states of the system (Equation (11)).

$$y = \left(\sum(HMM_{Observations} = Data_{Test})/n\right) * 100 \tag{11}$$

As the HMM works based on probabilities, 10,000 observation sequences were generated with a number equal to the test data, in order to calculate the respective accuracy and, later, to perform the average, in order to obtain the accuracy of the model in question, which was equal to 78.05%.

Having done the training of the model, it is already possible to obtain an HMM scheme for this case study (Figure 8), both with the transition probabilities between hidden states with the probability of emission from hidden states to observable states.

Now, once the parameters of the HMM model are known and embedded in the characteristic matrices, the conditions for using the "Viterbi" algorithm are met, which will allow the characterization of the most likely sequence of hidden states over time, which will give the diagnosing of the equipment's health.

Then, based on Figure 9, it is possible to see the evolution of the failure state of the drying press over time. To better understand the equipment diagnosis (Hidden States), the observable states extracted from the Clustering and the "raw" signal of each sensor is also represented in Figure 9.

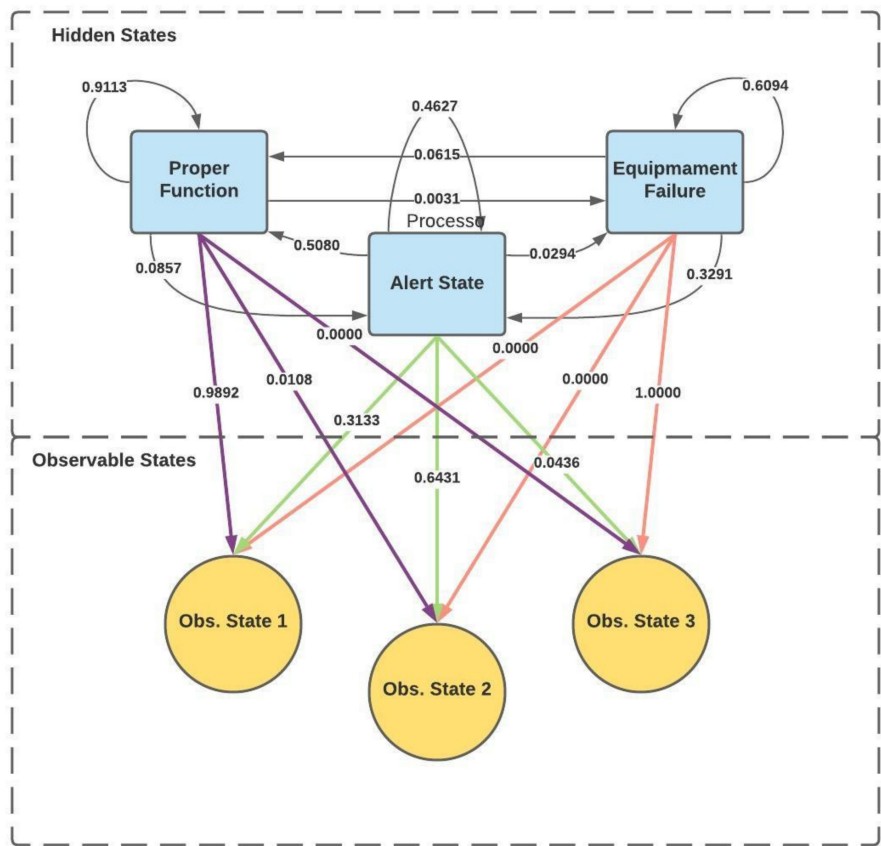

**Figure 8.** HMM scheme for transition and emission.

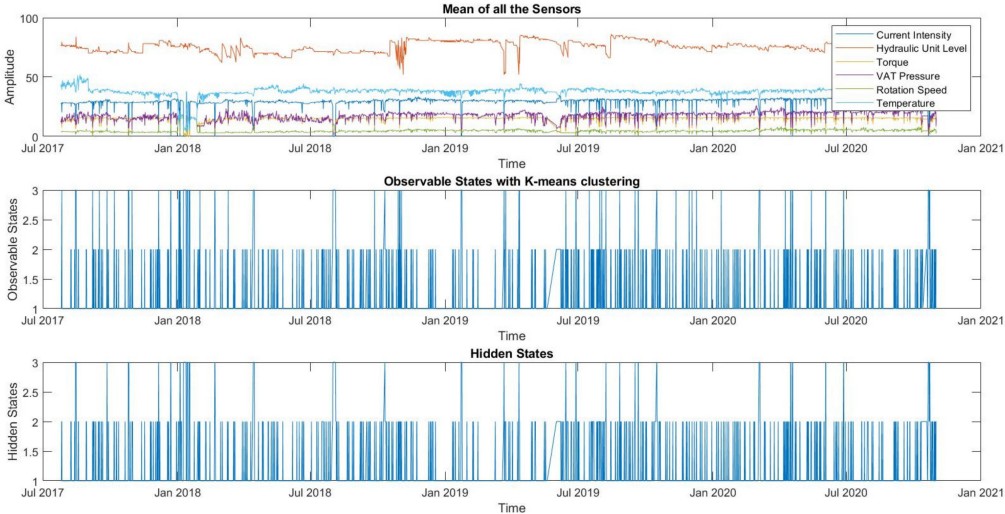

**Figure 9.** HMM Hidden States using the observable K-means Clustering.

## 5. Discussion

Observing the matrices, we evidenced, through the transition matrix, that hidden state 1 has a 91.13% probability of remaining in the same state and that it is unlikely to evolve to another state. Thus, as hidden state 1 represents the equipment in order of "good functioning", the ideal will always be to remain in this state. Hidden states 2 and 3, on the other hand, have some tendency to remain in the same state or to develop to the previous state, which can happen due to maintenance actions. Regarding the emission matrix, we found that the hidden state 1 occurs in a higher percentage when observing the observable state 1 and it never happens when observing the observable state 3. The opposite happens

with the hidden state 3. This makes sense because, as it was described along with this paper, observable state 1 was defined as the observable state that has the highest number of points, and by contrast, observable state 3 occurs the least number of times and, therefore, it is the one that is observed in the highest percentage, when the equipment is defective. Thus, hidden state 3 only occurs when observable state 3 is observed.

Apparently, the equipment had several alert states that return to a good working condition, in principle through (preventive) maintenance. As for the equipment failure status, this happened several times over the three years, which may signify that the drying press maintenance must be improved.

This paper tested several observable state "optimization" tools to insert into the HMM algorithm to perform the equipment diagnosis. As explained above, there is not a priori information about the equipment failure conditions being not possible, through the Working Orders (WO), to verify the effectiveness of the model. However, through graphical analysis, we can see, through Figure 9, that hidden state 3 happens in situations where the direct observations of the sensors deviate from the expected pattern.

Regarding the contribution of this model, it allows an online diagnosis of equipment from sensors, making possible the detection of failures without prior information about the equipment's condition. In addition, it is easy to adapt the number of features, depending on the equipment being sensed. Furthermore, the model can be generalized to any number of sensors and any equipment.

To improve the model in the future, we intend to make a diagnosis and prognosis of the equipment in an operation based on an online algorithm. For that, an Artificial Neural Network class, such as Multi-Layer Perceptron (MLP), may be used to improve the HMM performance. The objective of MLP is to predict which is the next observable state of the three observable states defined to, through the Viterbi algorithm, translating this observable state into a hidden state. Thus, it will be possible to say in advance what the health state of the equipment may be. Furthermore, to retrain the HMM model with the new observable states that may happen, a classification model, such as the Support Vector Machine (SVM), may be used, whose objective is to direct the new observable data read, for the respective cluster. In this way, through the new observable states that are happening, it is possible, through the Baum–Welch algorithm, to go retraining the HMM model. In addition, this study is expected to be extended to the company's five drying presses.

## 6. Conclusions

This paper presented a diagnostic method based on Hidden Markov Models.

The objective of the research described was to present an approach to diagnose failures in drying press equipment in a pulp industry, through HMM. The paper provides us with a theoretical basis for an application of a fault detection method through the optimization of observable states later applied to an HMM model. The added value of this failure detection method is due to the model making its prediction without prior information about the equipment conditions. The method was experimentally validated through a case study. For this, six variables were considered, through the collection of observations from six sensors, constituting a set of data with approximately three years and three months and with observations collected minute by minute.

The method was based on the optimization of observable states inserted in the HMM model, where data preparation is done first, followed by resource generation, where it generated 10 different statistical features, by which the objective was to better characterize the signals and do a reduction of the data. Then, a PCA is made and, finally, a Clustering, where each cluster represents the observable states to be used to train the HMM model. Then, a training and testing procedure was carried out on the HMM model to validate the method's ability to detect a failure.

Through the research carried out, it can be inferred that the method presented, based on HMM, can be used to diagnose the equipment health state supported on CBM strategies.

Based on this, companies have greater support for decision-making about equipment reliability, helping them be more competitive in the market.

Based on the previous case study, further research will be carried out with the aim of extending the time for equipment failure prediction as well as for Online Calibration Monitoring (OML). In this way, it will be possible to allow the extension or elimination of periodic sensor calibration intervals, allowing to improve the safety and reliability of observations through greater precision and reliability of the sensors used.

**Author Contributions:** Conceptualization, J.T.F., I.F., J.R. and A.J.M.C.; methodology, J.T.F. and I.F.; software, A.M. and I.F.; validation, J.T.F., I.F. and J.R.; formal analysis, J.T.F., I.F. and J.R.; investigation, A.M. and I.F.; resources, J.T.F., I.F., J.R. and A.J.M.C.; writing—original draft preparation, A.M.; writing—review and editing, J.T.F., I.F. and J.R.; project administration, J.T.F. and A.J.M.C.; funding acquisition, J.T.F. and A.J.M.C. All authors have read and agreed to the published version of the manuscript.

**Funding:** The research leading to these results has received funding from the European Union's Horizon 2020 research and innovation programme under the Marie Sklodowvska-Curie grant agreement 871284 project SSHARE and the European Regional Development Fund (ERDF) through the Operational Programme for Competitiveness and Internationalization (COMPETE 2020), under Project POCI-01-0145-FEDER-029494, and by National Funds through the FCT—Portuguese Foundation for Science and Technology, under Projects PTDC/EEI-EEE/29494/2017, UIDB/04131/2020, and UIDP/04131/2020.

**Conflicts of Interest:** The authors declare no conflict of interest.

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
