# Peer review of "Maintenance Prediction through Sensing Using Hidden Markov Models—A Case Study"

_applsci, doi:10.3390/app11167685_

Round 1
Reviewer 1 Report
The paper describes a case study, where from a set of six sensors an HMM is learned to detect the equipment health status. The study itself and the system are interesting, and presented in a structured way, however, there is room for improvement in the argument and decisions made, but not justified.
Overall, the system design to arrive at the HMM through several data pre-precessing steps makes sense. The selection of the individual methods for the steps is not well justified.
The data preparetions steps are rather rough, especcially replacing the system down time with mean value sensor readings seems crude, and is not justified beyond easier data handling.
The selection of features to compute from each of the sensors is just given, but not discussed in any way. Why are these 10 features chosen per sensor?
There are multiple other options for clustering methods, why is k-means chosen, and why others not?
In contrast, the standard methods employed, k-means, PCA, Z-transform, and HMM evaluation and training are described in depth. This seems unnecessary, since an abundance of literature is available on these, and they are used as is in the study.
Some smaller problems:
lines 329 ff, matrix B does not give the probability of a hidden state for an observable one, but the probability of an observation give a certain hodden state
section 4.5
Baum-Welch is an unsupervised training method, one can only specify the number of hidden states to be tranied, but NOT their meaning, this is a later interpretation and cannot be chosen beforehand, equating the least likely state with malfunction is educated guessing, even though it might be correct.
same section: how was the data divided into 70% train and 30% test? randomly, by time, or by what? A bad decision here can lead to a less valid model, so it needs to be documented
The validation step in equation 6 seems strange, please provide a source, where this process is described.
In the discussion, line 602ff, the equipment downtimes were replaced, so there is no way to find out how long the maintenance lasted, and the conclusion seems not valid
Reviewer 2 Report
In this research work is presented a condition monitoring and fault diagnosis based on Hidden Markov Models, the proposed approach is applied to identify three different working conditions in an industria process.
The manuscript have to be improved since there are several issues that must be attended.
-The current organization and structure of the manuscript must be corrected, the authors must follow the structure of the "ApplSci" template.
-In the current Section 1 "Introduction", are described several facts that are important to perform the proposed research, however, such facts no not present or have an order.
-In most of the research papers are included a specific section to provide all mathematical and theoretical basis, and this section is commonly presented previous to the Section where "The proposed method is presented"
-The flow chart of the proposed method is presented in a general way, a more detailed flow chart must be included; additionally, the description of the proposed method must be improved.
-The feature calculation or signal characterization is one of the most important stages of proposed condition monitoring strategies, in this proposal such feature calculation is based on the calculation of a statistical set of features. Why statistical time domain-based features are considered in this work? does the statistical set of features provide sufficient and importan information to achieve the fault identification? The corresponding equations of the considered set of features may be appended in the manuscript or at least must be referenced to other published works (i.e. https://doi.org/10.1016/j.measurement.2021.109404)
-The quality of Figures must be improved, i.e. the values of vertical axis in Figure 6 appear overlapped.
-The discussion of the obtained results may be improved in order to highlight the contribution and novelty of the proposed work; also, it must be mentioned what are the advantages of the proposed in regard to other related works.
Round 2
Reviewer 2 Report
The authors have attended all the made suggestions.
The contribution, novelty and readability of the manuscript has been improved significantly.